# Sleep Habits, Academic Performance and Health Behaviors of Adolescents in Southern Greece

**DOI:** 10.3390/healthcare12070775

**Published:** 2024-04-03

**Authors:** Christina Alexopoulou, Maria Fountoulaki, Antigone Papavasileiou, Eumorfia Kondili

**Affiliations:** 1Department of Intensive Care and Sleep Laboratory, University Hospital of Heraklion, 71110 Heraklion, Greece; kondylie@uoc.gr; 2Attikon University Hospital, 12462 Athens, Greece; mafountoulaki@gmail.com; 3Pediatric Neurology, IASO Children Hospital Athens, 15123 Marousi, Greece; theon@otenet.gr

**Keywords:** sleep, sleep hygiene, adolescents, sleep habits, sleep deprivation, academic performance, health behavior

## Abstract

Adolescents often experience insufficient sleep and have unhealthy sleep habits. Our aim was to investigate the sleep patterns of secondary education students in Heraklion, Crete, Greece and their association with school performance and health habits. We conducted a community-based cross-sectional study with 831 students aged 13–19 years who completed an online self-reported questionnaire related to sleep and health habits. The data are mostly numerical or categorical, and an analysis was performed using *t*-tests, chi-square tests and multiple logistic regression. During weekdays, the students slept for an average of 7 ± 1.1 h, which is significantly lower than the 7.8 ± 1.5 h average on weekends (*p* < 0.001). Nearly 79% reported difficulty waking up and having insufficient sleep time, while 73.8% felt sleepy at school at least once a week. Having sufficient sleep time ≥ 8 h) was positively correlated with better academic performance (OR: 1.48, CI: 1.06–2.07, *p* = 0.022) and frequent physical exercise (never/rarely: 13.5%, sometimes: 21.2%, often: 65.3%; *p* = 0.002). Conversely, there was a negative correlation between adequate sleep and both smoking (OR: 0.29, CI: 0.13–0.63) and alcohol consumption (OR: 0.51, CI: 0.36–0.71, *p* = 0.001). In conclusion, this study shows that students in Heraklion, Crete frequently experience sleep deprivation, which is associated with compromised academic performance, reduced physical activity and an increased likelihood of engaging in unhealthy behaviors like smoking and alcohol consumption.

## 1. Introduction

Sleep behavior among adolescents is a major health concern worldwide [1]. When discussing adolescent sleep behavior, it is important to consider various key points such as the onset of sleep time, sleep patterns and duration and how they correlate with gender, the day of the week, parental sleep behaviors and other social factors. Studies have shown that the quality of sleep can be negatively affected depending on these factors [2,3].

Various studies have reported that the timing of sleep onset differs according to age, particularly in adolescents. Specifically, older adolescents tend to fall asleep later on both school nights and weekends [1,2,4,5,6]. However, they tend to sleep longer on weekends than on school nights. School demands often disrupt adolescents’ sleep–wake cycles, leading to inconsistent sleep patterns. They often become deprived of sleep during the week and make up for it by sleeping more during the weekends. However, more research is required to determine the appropriate sleep duration for young patients aged 13–19. Previous reports have suggested that adolescents need approximately 9 h of sleep per night on average [6,7]. Nevertheless, this differs significantly from the actual time they spend sleeping, which decreases with each passing decade in some areas of the world. The teenage population in the US and Sweden tends to have a shorter sleep duration in later decades compared to past decades, although the average sleep requirement for all adolescents is considered to be 9 h [1,2,4,5,6]. North American, European, Chinese and Australian studies have revealed that older adolescents (15–16 years) have a borderline sleep onset time (in order to achieve the full 9 h sleep they need), while Icelandic adolescents of all ages go to sleep later than others (borderline to insufficient) [1]. According to Gariepi et al. [2], the sleep recommendation times were, on average, lower on school days than on non-school days with great variation among countries, especially on school days and between genders.

Additionally, the onset of puberty varies considerably in an individualized fashion, and some adolescents are still prepubertal when they enter middle school. It should be noted that the sleep needs in the prepubertal pediatric population are not yet known [8]. 

Differences in sleep duration between boys and girls have been described, with some studies pointing out that boys sleep less and others concluding that they sleep more than girls [4,5,6,9]. 

The quality of sleep has been linked to academic performance and social jet lag [8,10]. Sleep deprivation, characterized by insomnia and daytime sleepiness, can have negative effects on adolescents’ school performance as well as their somatic and psychosocial health [8]. Sleep behavior is a prime example of how external factors like parental bedtime setting, cognitive behavioral therapy for insomnia (CBT-I) and a practice of good ‘sleep hygiene’ can modify children’s and adolescents’ physiological and developmental needs, leading to important outcomes for cognition, emotion regulation, motivation and mental health [11,12,13,14]. Environmental factors can vary between countries and regions, highlighting the need for further studies on sleep behaviors in different populations.

To the authors’ knowledge, there is no study in the literature examining sleep habits, academic performances and health behaviors and the correlation between them in adolescents in Greece, so we conducted this study to assess both the duration and quality of sleep, examining factors such as bedtime, wake-up time, variations between school days and weekends and experiences of difficulties in waking up or daytime sleepiness. Additionally, we sought to investigate any correlation between sleep patterns and various aspects such as academic performance and health behaviors, including smoking, the consumption of coffee and alcohol intake as well as the use of electronic devices.

## 2. Materials and Methods

### 2.1. Participants and Methods

This was a cross-sectional, community-based study with a self-reported questionnaire. The participants were school students during their sixth years of secondary education, separated into two groups corresponding to the first part of 3 years of Gymnasium and the second part of 3 years of Lyceum each, in accordance with the Greek educational system. This study was conducted in 2018 among students aged between 13 and 19 years in Crete, Southern Greece before the emergence of the COVID-19 pandemic. This study was approved by the Ethics Committee of the University Hospital of Heraklion, Crete, Greece. One of the teachers in each class was responsible for informing the parents and students about the study and obtaining parental consent. All parents consented and all students who were recruited participated in the study.

As part of the study, 831 students aged between 13 and 19 were asked to anonymously complete an online 28-question questionnaire. The questions covered their everyday habits, including sleep duration and bedtime, sleepiness in class, difficulties in waking up, sleep hygiene and habits before bed, such as the use of electronic devices. Additionally, the students were asked to record their coffee and alcohol intake, exercise routines and school performances. They were also asked about their parents’ sleep habits because it seems that there is a correlation. The questionnaire was accessible through http://j.mp/sleep1718 accessed on 15 January 2018 (Table 1).

### 2.2. Statistical Analysis

Categorical variables were expressed as frequencies and percentages. Continuous variables were presented as mean ± standard deviation (SD) or median and 25–75% interquartile range (25–75 IQR), as appropriate. Categorical variables are compared using the Fisher exact test, and continuous variables were compared using the Kruskal–Wallis, Friedman and Wilcoxon or Mann–Whitney tests, as appropriate. Spearman’s rho was used to evaluate correlations between continuous and categorical variables, as appropriate. To investigate if sleep habits and other variables were independently associated with poor academic performance, a multiple logistic regression model (with odds ratios [ORs] and two-sided 95% confidence intervals [Cis]) was performed. A *p*-value of < 0.05 was considered statistically significant. Pairwise comparisons were performed using the Kruskal–Wallis test followed by Bonferroni’s correction for multiple comparisons. The adjusted significance level was set at 0.001. Statistical analyses were performed using an IBM SPSS Statistics 24.0 statistical package.

## 3. Results

A total of 831 students completed the questionnaires. The age and gender distributions at Gymnasium and Lyceum and class levels are listed in Table 2. Most of the participants (70.2%) were Lyceum students. There were no significant age and gender differences within the education and class levels (*p* > 0.05).

### 3.1. Sleep Habits

We observed that, regardless of the level of education, the students slept for a significantly shorter duration on weekdays (median 7.0 h, with an interquartile range (IQR) of 6–7 h) compared to weekends (median 8.0 h, IQR 7–9 h, *p* < 0.001), as shown in Figure 1. A similar pattern was observed in the sleep durations of the students’ parents, with shorter sleep durations on weekdays (median 7.0 h, IQR 6–8 h) compared to weekends (median 8.0 h, IQR 7–8 h), as shown in Table 3. The distributions of students and students’ parents at six different hours of sleep duration (4–9 h) are presented in Figure 2. The sleep durations in the six different classes during weekdays and weekends are presented in Figure 3 and Table 4. Overall, a significant decrease in sleep duration with an increase in educational year was observed. The sleep duration decreased from 8 (IQR 7–8) hours in the first Gymnasium class to 7 (6–7) hours in the third Lyceum class, indicating a decline in sleep duration with the progression of educational level (*p* < 0.001). Pairwise comparisons of sleep during weekdays and weekends between the six classes are detailed in Appendix A.

A significant bedtime delay was observed on weekdays as the students progressed to higher educational levels. The bedtime was reported to be 22:30 for the first class of Gymnasium, 23:30 for the third class of Gymnasium and 00:10 for the third class of Lyceum, (*p* < 0.001). Bedtime delays during weekends followed a similar pattern across different educational levels, with bedtimes shifting later as the educational levels increased, being 23:54 in the first Gymnasium class and 1.06 in the third Lyceum class. Weekends showed a consistent trend of delayed bedtimes compared to weekdays across all educational levels (*p* < 0.001). 

Of the 831 students surveyed, 193 (23.2%) reported adhering to the recommended sleep durations for their ages (equal to or exceeding 8 h). A total of 250 students (19.7%) disclosed having at least one nap per week, whereas 86 students (10.3%) admitted to having more than two nap days per week. No statistically significant differences in nap patterns were observed between the male and female students.

Regarding pre-sleep habits involving electronic devices, a substantial portion of the surveyed students—529 individuals (63.7%)—acknowledged using electronic devices before bedtime. We found a difference between genders in regard to using electronic devices before sleep, with a significantly higher rate observed among boys compared to girls; 69.6% of boys reported using electronic devices before sleep compared to 57.6% of girls (*p* < 0.001). 

### 3.2. Health Habits

The students’ health habits, encompassing behaviors related to smoking, alcohol consumption and physical activity, were examined and are presented in Table 5. Among the surveyed students, 25 individuals (3.0%) admitted to frequent smoking, while 56 students (6.7%) reported occasional smoking. We found a significant gender disparity in smoking frequency: 5.0% of boys acknowledged frequent smoking, and 8.3% disclosed occasional smoking. In comparison, only 1.0% of girls reported frequent smoking, and 5.1% admitted to occasional smoking (*p* < 0.001).

For alcohol consumption, occasional drinking “drinking alcohol sometimes” was reported by 373 participants (44%), with a significantly higher prevalence among boys (207, 49.2%) compared to girls (166, 40.5%) (*p* < 0.001). A relatively uniform prevalence of coffee and caffeine consumption was found between boys and girls. The consumption of coffee and caffeine-containing products was reported among 115 students (13.8%), and no significant difference was found among both genders. A majority of students (54.3%) reported exercising often, with boys exercising at a higher proportion (58.4%) than girls (50.0%) (*p* = 0.046). Health habits per age are presented in Appendix A. The percentage of students who reported never or rarely smoking decreased significantly with age, starting at 97% at the age of 13 and reaching 81.5% and 42.9% at 18 and 19, respectively. A similar pattern was observed for drinking alcohol (89.7% at the age of 13 and 34.2% and 27.8% at the ages of 18 and 19, respectively) and consuming coffee and caffeine products (77.8% at the age of 13 and 33.8% and 28.6% at the ages of 18 and 19, respectively).

Regarding physical activity, we found an increase in the proportion of students who reported never or rarely engaging in physical activity with age, from 8.5% at the age of 13 to 33.4% and 57.1% at the ages of 18 and 19, respectively. Furthermore, there was a significant decrease in the proportion of students who reported engaging in physical activity often as they became older, from 76.9% at the age of 13 to 26.5% and 28.6% at the ages of 18 and 19, respectively.

### 3.3. Sleep Quality

The evaluation of sleep quality centered on difficulties in waking up and experiencing daytime sleepiness. Among the surveyed students, a substantial majority, comprising 656 individuals (78.9%), reported difficulties when waking up. Within this group, 292 students (35.1%) encountered these difficulties often. A total of 304 students (36.6%) admitted to feeling sleepy at school for one to two days per week, while 309 students (37.2%) reported feeling the same for over two days per week. Significantly more boys than girls were found to never or rarely feel sleepy at school (29.7% vs. 22.7%, *p* < 0.05). 

A significant positive correlation emerged between sufficient sleep duration, defined as a sleep duration of ≥8 h, and academic performance (18.1 vs. < 18.0), as indicated by the school grades (OR: 1.48 CI 1.06–2.07, *p* = 0.022) after adjusting gender and educational level (Lyceum/Gymnasium) (Appendix A). Additionally, sufficient sleep duration exhibited a positive association with the frequency of engaging in physical exercise, with higher rates observed among those who exercised often (65.3%) compared to those who exercised sometimes (21.2%) or rarely (13.5%) (*p* = 0.002). Conversely, sufficient sleep demonstrated a negative correlation with both smoking (OR: 0.29, CI 0.13–0.63) and alcohol consumption (OR: 0.51, CI 0.36–0.71) (*p* = 0.001). 

## 4. Discussion

The results of this cross-sectional study reveal that a significant proportion of adolescents in Crete, Greece fail to attain good sleep health. Specifically, this study shows that most adolescents are unable to achieve adequate sleep duration, appropriate timing and sustained alertness during waking hours. Furthermore, this study identifies a correlation between insufficient sleep time, lower academic performance and an increased prevalence of unhealthy habits. The current study found that the total sleep time was lower than that considered appropriate for each age group. According to the literature, less than 8 h of sleep is not appropriate for optimal health for individuals aged between 13 and 18 [7,10]. Sleep duration decreased from 8 (IQR 7–8) hours in the first Gymnasium class to 7 (6–7) hours in the third Lyceum class, indicating a decline in sleep duration with the progression of educational level (*p* < 0.001). Sleep duration was increased during weekends but still not sufficient. Previous studies reported inconsistent findings regarding gender-based disparities in sleep quality among adolescents, including daytime sleepiness, difficulty initiating sleep and sleep onset latency [1,11,15,16,17,18,19]. However, the present study did not find significant differences in sleep quality indices between genders, with the exception of boys reporting lower levels of daytime sleepiness compared to girls. Regardless of their education level (Gymnasium or Lyceum), we found that students have shorter sleep durations on weekdays than on weekends. The same pattern was also observed in the students’ parents. Furthermore, with the progression of the academic level, there was a noticeable reduction in sleep duration on both weekdays and weekends. This finding aligns with the consensus statement of the American Academy of Sleep Medicine in 2016 (7) indicating a sleep phase delay as children move through secondary school years. Notably, adolescents in Lyceum tend to sleep less than 8 h per night, which is significantly less than the recommended time for their age [7]. During the transition from Gymnasium to Lyceum, students are exposed to various environmental demands, such as school demands and socialization, which may affect their sleep quantity. This period is particularly risky as students are expected to balance their educational and extracurricular activities, leading to inadequate sleep. Insufficient sleep during this critical period may have negative impacts on students’ mental and physical health, academic performance and overall well-being [20].

A considerable proportion of students reported taking at least one nap per week. Remarkably, there was no apparent disparity in napping patterns between genders. However, it is difficult to interpret the significance of this finding, as napping needs have been linked to factors such as insufficient sleep at night or ethnic, socio-cultural and racial differences [8]. Nevertheless, this finding emphasizes the need for further research studies in specific geographical regions. Although young individuals are unlikely to require different sleep durations across different continents, they may have diverse sleep patterns that are influenced by environmental factors such as warm climates or socio-cultural norms.

Using electronic devices before sleep was reported by the majority of the students, with a significantly higher rate in boys than girls. This finding holds significant importance as such behavior has been linked to delayed sleep onset. Prior research has consistently demonstrated that the use of electronic devices before bedtime has detrimental effects on sleep quality, including delayed onset, reduced duration and disrupted sleep patterns [21,22,23].

According to this study’s results, a considerable number of participants experienced poor sleep quality, as evidenced by difficulties waking up and feeling sleepy during the day. This finding is significant since inadequate sleep during adolescence has been linked to cognitive dysfunction, learning challenges, poor emotional well-being and metabolic disorders that can lead to obesity [24,25,26,27,28,29,30,31]. 

The present study has revealed a positive correlation between sleep duration and physical exercise frequency. Specifically, children who reported experiencing good sleep quality were found to engage in physical exercise significantly more frequently than their peers who exercised seldomly or not at all. This finding is in accordance with the study by Nixon et al. [32], who reported that adolescents with poor sleep quality might engage in less physical activity. Both studies suggest that the quality and quantity of sleep are important factors in determining physical activity levels and overall health in children and emphasize the importance of quality sleep and its impact on children’s physical well-being.

In the current study, we found that good sleep is linked with lower rates of smoking and alcohol consumption. This finding is consistent with previous studies that have shown a connection between poor sleep and risky behaviors in adolescents [33,34,35,36]. A national sample of US adolescents revealed that those with late bedtimes had an increased likelihood of using cigarettes, alcohol and drugs. They were also more prone to violent behaviors and emotional distress [37]. Furthermore, the impact of sleep quality on cognitive function has been highlighted in healthy adolescents and those with neurodevelopmental and psychiatric conditions [38]. Previous studies have demonstrated that sleep quality is critically involved in learning and memory [11].

A significant finding of this study is that sufficient sleep duration, defined as at least eight hours of sleep daily, was positively related to academic performance, as assessed by school grades. This finding is in accordance with previous studies reporting that adequate sleep quality is associated with higher academic performance [39]. The association between good sleep quality and high academic performance has been extensively investigated, and multiple possible factors have been proposed [40]. Sleep is crucial for cognitive functions like attention, memory and decision making. Adequate sleep can enhance these functions, enabling adolescents to concentrate better in class, retain information effectively and perform well in academic tasks. Moreover, good sleep quality is essential for memory consolidation and enhancing learning. Sufficient sleep may promote memory and recall information processes, leading to increased learning capacity and, ultimately, high academic performance [41,42].

Addressing and improving sleep hygiene, promoting consistent sleep schedules, creating a conducive sleep environment and raising awareness about the importance of adequate sleep among adolescents and their parents can significantly mitigate these effects and potentially improve academic outcomes.

### Limitations 

The limitation of this study is the reliance upon retrospective data to measure sleep instead of employing sleep diaries, which is regarded as the most reliable method for subjectively assessing sleep. Sleep diaries are easy to use, transportable and cost-effective. They measure sleep proactively, thereby eliminating any bias attributed to the tendency to recollect recent experiences [43]. 

The questionnaire used was made on the Google platform for the needs of this study and was not a validated scale that already existed in the literature. The aim was to cover all of the aspects of interest without using many different scales, which would eliminate the answering rates among school students due to long answering times. It is of great interest that the participation rate among the students was almost 100%. All health-associated behaviors were self-reported, and the data on alcohol and smoking were crude. As a measure of academic achievement, we only used the school grades at the moment of participation in the study, which may not reflect the overall academic performance of a student. This study was conducted on Crete, which is the biggest island in Greece, but it is a specific region and the generalization of the results is limited. Further studies are needed to better evaluate the sleep habits and health-related behaviors among adolescents as well as the effects of specific interventions in order to improve their sleep and well-being.

## 5. Conclusions

In conclusion, this cross-sectional study showed that most Greek adolescents residing on Crete experience poor sleep quality and sleep deprivation. As the adolescents grew older, their sleep duration tended to decrease, and no distinctions were observed between genders in this regard within this sample. More importantly, this research highlighted a positive correlation between good sleep habits and both academic achievement and the avoidance of early risky behaviors among these adolescents. These findings have implications for academic and healthcare professionals working with adolescents in Greece and beyond. Promoting good sleep hygiene may be an effective strategy for improving academic performance and reducing risky behaviors in this population. Further research is warranted to gain a more comprehensive understanding of the factors that influence sleep behaviors in adolescents.

## Figures and Tables

**Figure 1 healthcare-12-00775-f001:**
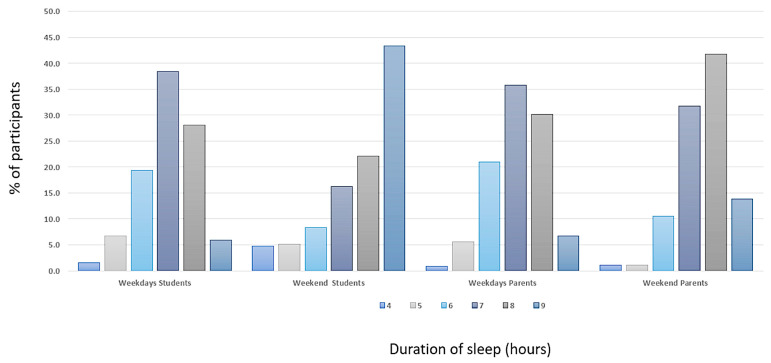
Percentages of students and students’ parents at 5 different ranges of sleep duration during weekdays and weekends.

**Figure 2 healthcare-12-00775-f002:**
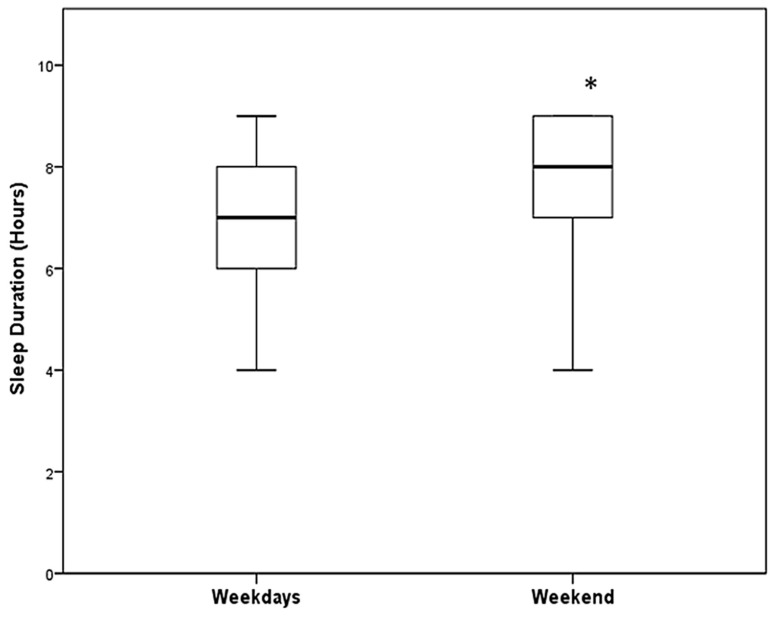
Box and whisker plots representing a comparison of sleep duration on weekdays and weekends for students./ The lower and upper edges of the box represent the 25th and 75th percentiles, respectively. The lines within the box show the median values. The whiskers depict the adjacent values. (*) denotes a statistically significant difference between the values; *p* < 0.05.

**Figure 3 healthcare-12-00775-f003:**
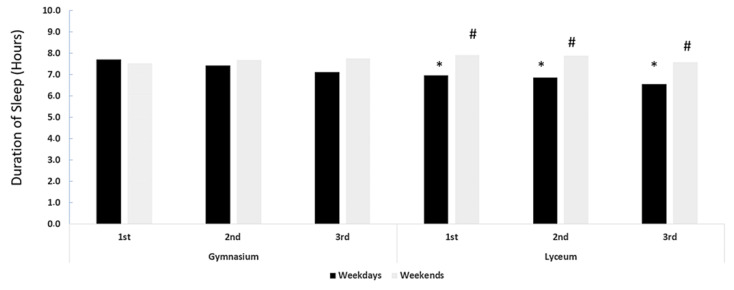
Sleep duration during weekdays (black columns) and weekend days (light grey columns) for each school class. Pairwise comparisons performed using Kruskal–Wallis test, followed by Bonferroni’s correction for multiple comparisons. * represents significant difference from Gymnasium 1st class for weekdays, *p* < 0.0001, while # represents significant difference from Gymnasium 1st class for weekends.

**Table 1 healthcare-12-00775-t001:** Questionnaire (translated to English).

Sleep in Adolescent Students					
A1. Sex					
Female			Male		
A2. Class					
1st Gymnasium		2nd Gymnasium		3rd Gymnasium	
1st Lyceum		2nd Lyceum		3rd Lyceum	
A3. Year of birth					
1999		2000		2001	
2002		2003		2004	
2005					
A4. Your grades (without gymnastics) on the first quartermaster this year. If you haven’t received your grades yet please select your estimation. (max 20)					
18.1–20		16.1–18		13.1–16	
9.5–13		>9.5			
A5. Do you exercise?					
Never		Sometimes		On a regular often basis	
A6. Do you smoke?					
Not at all		Sometimes		Almost every day	
A7. Do you drink any alcohol?					
Not at all		Sometimes		Almost every day	
**Sleep Duration at Home**					
Β1. How many hours do you sleep the nights before school?					
4 h or less	about 4.5 h	about 5 h	about 5.5 h	about 6 h	about 6.5 h
about 7 h	about 7.5 h	about 8 h	about 8.5 h	about 9 h	about 9.5 h
>9 h					
Β2. The same days how many hours do your parents sleep?					
4 h or less	about 4.5 h	about 5 h	about 5.5 h	about 6 h	about 6.5 h
about 7 h	about 7.5 h	about 8 h	about 8.5 h	about 9 h	about 9.5 h
>9 h					
Β3. During Friday Saturday and holiday eves, when you don’t have to go to school the next morning, how many hours do you sleep?					
4 h or less	about 4.5 h	about 5 h	about 5.5 h	about 6 h	about 6.5 h
about 7 h	about 7.5 h	about 8 h	about 8.5 h	about 9 h	about 9.5 h
>9 h					
Β4. The same days how many hours do your parents sleep?					
4 h or less	about 4.5 h	about 5 h	about 5.5 h	about 6 h	about 6.5 h
about 7 h	about 7.5 h	about 8 h	about 8.5 h	about 9 h	about 9.5 h
>9 h					
Β5. During holiday season how many hours do you sleep?					
4 h or less	about 4.5 h	about 5 h	about 5.5 h	about 6 h	about 6.5 h
about 7 h	about 7.5 h	about 8 h	about 8.5 h	about 9 h	about 9.5 h
>9 h					
Β6. The same days how many hours do your parents sleep?					
4 h or less	about 4.5 h	about 5 h	about 5.5 h	about 6 h	about 6.5 h
about 7 h	about 7.5 h	about 8 h	about 8.5 h	about 9 h	about 9.5 h
>9 h					
**About bed time at home**					
C1. At what time do you sleep the nights before school?					
Around 8:30 p.m. or earlier		Around 09:00 p.m.		Around 09:30 p.m.	
Around 10:00 p.m.		Around 10:30 p.m.		Around 11:00 p.m.	
Around 11:30 p.m.		Around midnight		Around 12:30 a.m.	
Around 1:00 a.m		Around 01: 30 a.m.		Later than 01:30 a.m.	
C2. The same days at what time do your parents sleep?					
Around 8:30 p.m. or earlier		Around 09:00 p.m.		Around 09:30 p.m.	
Around 10:00 p.m.		Around 10:30 p.m.		Around 11:00 p.m.	
Around 11:30 p.m.		Around midnight		Around 12:30 a.m.	
Around 1:00 a.m.		Around 01:30 a.m.		Later than 01:30 a.m.	
C3. During Friday Saturday and holiday eves, when you don’t have to go to school the next morning, at what time do you sleep?					
Around 8:30 pm or earlier		Around 09:00 p.m.		Around 09:30 p.m.	
Around 10:00 p.m		Around 10:30 p.m.		Around 11:00 p.m.	
Around 11:30 p.m.		Around midnight		Around 12:30 a.m.	
Around 1:00 a.m.		Around 01:30 a.m.		Later than 01:30 a.m.	
C4. At the same days at what time do your parents sleep?					
Around 8:30 p.m. or earlier		Around 09:00 p.m.		Around 09:30 p.m.	
Around 10:00 p.m.		Around 10:30 p.m.		Around 11:00 p.m.	
Around 11:30 p.m.		Around midnight		Around 12:30 a.m.	
Around 1:00 a.m.		Around 01:30 a.m.		Later than 01:30 a.m.	
C5. During holiday season at what time do you sleep?					
Around 8:30 pm or earlier		Around 09:00 p.m.		Around 09:30 p.m.	
Around 10:00 p.m.		Around 10:30 p.m.		Around 11:00 p.m.	
Around 11:30 p.m.		Around midnight		Around 12:30 a.m.	
Around 1:00 a.m.		Around 01: 30 a.m.		Later than 01:30 a.m.	
C6. At the same days at what time do your parents sleep?					
Around 8:30 p.m. or earlier		Around 09:00 p.m.		Around 09:30 p.m.	
Around 10:00 p.m.		Around 10:30 p.m.		Around 11:00 p.m.	
Around 11:30 p.m.		Around midnight		Around 12:30 a.m.	
Around 1:00 a.m.		Around 01:30 a.m.		Later than 01:30 a.m.	
**Everyday sleep and wake**					
D1. Do you have any difficulties to wake up in the morning during school days?					
Never or rarely		Sometimes		Almost always	
D2. How do you go to school usually?					
On foot			By car or bus		
D3. What time do you usually leave from home to go to school?					
.....................................................................................................................................................................................					
D4. During the daytime at school do your feel sleepy?					
Never or rarely		1–2 days/week		>2 days/week	
D5. Do your parents believe that you need more sleep during schooldays and do they advise you to sleep more or earlier?					
Never or rarely		Sometimes		Almost always	
Not only during school days but everyday					
D6. During schooldays do you take a nap?					
Never or rarely		1–2 naps per week		>2 naps per week	
D7. During the school year your everyday afterschool activities (sports, extra lessons etc.) stop at:					
7:00 p.m.	8:00 p.m.	9:00 p.m.	10:00 p.m.	Later than 10:00 p.m.	
D8. During the school year what is the latest thing you do before sleep?					
Watch TV	Use PC or smartphone	Read a book	Listen Music	Play without electronic means	Homework
Else:.........................................................................................................................................................					
D9. During school year do you take any caffeine products or energy drinks or substances in order to stay awake?					
Never or rarely		Sometimes		Almost everyday	

**Table 2 healthcare-12-00775-t002:** Participants’ characteristics. Distributions based on gender, class and educational level.

	Sex									
	Boys (n = 421)				Girls (n = 410)			Total (n = 831)		
		n	% ^#^	% *	N	% ^#^	% *	n	% *	*p*
Age (years)	13	57	48.7	13.5	60	51.3	14.6	117	14.1	0.434
	14	44	50.0	10.5	44	50.0	10.7	88	10.6	
	15	21	44.7	5.0	26	55.3	6.3	47	5.7	
	16	130	57.0	30.9	98	43.0	23.9	228	27.4	
	17	96	49.7	22.8	97	50.3	23.7	193	23.2	
	18	70	46.4	16.6	81	53.6	19.8	151	18.2	
	19	3	42.9	0.7	4	57.1	1.0	7	0.8	
Educational Level	Gymnasium	120	48.4	28.5	128	51.6	31.2	248	29.8	0.392
	Lyceum	301	51.6	71.5	282	48.4	68.8	583	70.2	
Classs	Gymnasium 1st	56	48.7	13.3	59	51.3	14.4	115	13.8	0.316
	Gymnasium 2nd	44	50.6	10.5	43	49.4	10.5	87	10.5	
	Gymnasium 3rd	20	43.5	4.8	26	56.5	6.3	46	5.5	
	Lyceum 1st	135	57.0	32.1	102	43.0	24.9	237	28.5	
	Lyceum 2nd	93	48.2	22.1	100	51.8	24.4	193	23.2	
	Lyceum 3rd	73	47.7	17.3	80	52.3	19.5	153	18.4	

#: percentage per gender for the same class; *: percentage of the total students in the same class.

**Table 3 healthcare-12-00775-t003:** Sleep durations on weekdays and weekends for students and students’ parents.

Sleep Duration (Hours) (Median, IQR 25–75)	Weekdays	Weekends	*p*-Value
Students	7 (6–8)	8 (7–9)	<0.001
Students’ Parents	7 (6–8)	8 (7–8)	<0.001

**Table 4 healthcare-12-00775-t004:** Sleep durations based on class on weekdays and weekends.

Sleep Duration (Hours), Median (IQR 25–75)	Class					
	Gymnasium 1st	Gymnasium 2nd	Gymnasium 3rd	Lyceum 1st	Lyceum 2nd	Lyceum 3rd
Weekdays	8 (7–8)	8 (7–8)	7 (7–8)	7 (6–8)	7 (6–7)	7 (6–7)
Weekends	8 (7–9)	8 (7–9)	8 (7–9)	9 (7–9)	8 (7–9)	8 (7–9)

**Table 5 healthcare-12-00775-t005:** Health habits (smoking, drinking and exercise) of boys and girls.

		Sex									
		Boy			Girl			Total			
		N	% ^1^	% ^2^	N	% ^1^	% ^2^	N	% ^1^	% ^2^	*p*-Value
Smoking	Never/Rarely	365	48.7	86.7	385	51.3	93.9	750	100.0	90.3	<0.001
	Sometimes	35	62.5	8.3	21	37.5	5.1	56	100.0	6.7	
	Often	21	84.0	5.0	4	16.0	1.0	25	100.0	3.0	
	Total (per gender)	421			410						
Drinking alcohol	Never/Rarely	202	45.7	48.0	240	54.3	58.5	442	100.0	53.2	<0.001
	Sometimes	207	55.5	49.2	166	44.5	40.5	373	100.0	44.9	
	Often	12	75.0	2.9	4	25.0	1.0	16	100.0	1.9	
	Total (per gender)	421			410						
Exercise	Never/Rarely	73	47.4	17.3	81	52.6	19.8	154	100.0	18.5	0.046
	Sometimes	102	45.1	24.2	124	54.9	30.2	226	100.0	27.2	
	Often	246	54.5	58.4	205	45.5	50.0	451	100.0	54.3	
	Total (per gender)	421			410						
Coffee or	Never/Rarely	253	51.5	60.1	238	48.5	58.0	491	100.0	59.1	0.821
caffeine	Sometimes	112	49.8	26.6	113	50.2	27.6	225	100.0	27.1	
	Often	56	48.7	13.3	59	51.3	14.4	115	100.0	13.8	
	Total (per gender)	421			410						

^1^: Percentage of total number of participants at each frequency condition (Never/Rarely, Sometimes and Often); ^2^: percentage of total number of participants within same gender at each frequency condition; *p* < 0.001 represents statistically significant difference.

## Data Availability

Data are contained within the article.

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
