# Peer review of "Sleep Habits, Academic Performance and Health Behaviors of Adolescents in Southern Greece"

_healthcare, 2024, doi:10.3390/healthcare12070775_

Round 1

Reviewer 1 Report

Comments and Suggestions for Authors

Abstract

Since in the results other health behaviours are listed, the authors should also mention earlier in the abstract that along with sleep other health behaviours were also assessed.

Introduction The introduction is clear albeit a bit brief.

This sentence (p. 2 lines 49-50) could be refined as it is vague and not well linked with this paragraph – what is meant by external factors? : Sleep behavior is a prime example of how external factors can modify children's and adolescents' physiological and developmental needs, leading to important outcomes for cognition, emotion regulation, motivation, and mental health [10].

Method

In the supplementary document there are question about parents’ sleep, but this is not mentioned in the method section. Please give rationale why this was measured. What was the basis for all these questions posed to students? Which sleep studies?

The measure of academic achievement was very crude and subjective. This needs to be highlighted in the limitations of the study.

Results

I do not think it is precise or correct to call social habits behaviours like drinking or smoking. Please change it as it is very misleading. The same applies to the Discussion.

What is the assocation between sleep duration in the week and at weekend?

In the section on Sleep quality a figure x is mentioned in the text. I assume it is a mistake?

Discussion

Page 7 – what is meant  by environmental demands? In this context authors mean, I think, school demands and socialising. Please refine it.

Can napping be more common in country like Greece?

Please elaborate more on the link between parents' and their children sleep.

Please elaborate on the limitations of your measures of all behaviours - it was self-report, plus measures of alcohol and smoking were very crude. Why wasn't any validated health questionnaire used in the study? 

Comments on the Quality of English Language

Minor editing of English language required.

Author Response

Response to Reviewer 1

Thank you very much for taking the time to review this manuscript. Please find the detailed point to point response below and the corresponding revisions/corrections highlighted in the re-submitted files.

Comment

Abstract

Since in the results other health behaviors are listed, the authors should also mention earlier in the abstract that along with sleep other health behaviors were also assessed.

Response: Thank you for pointing this out. We agree with this comment. Therefore, we have made changes in the text.

Comment

This sentence (p. 2 lines 49-50) could be refined as it is vague and not well linked with this paragraph – what is meant by external factors? : Sleep behavior is a prime example of how external factors can modify children's and adolescents' physiological and developmental needs, leading to important outcomes for cognition, emotion regulation, motivation, and mental health [10].

Response: Thank you for pointing this out. We agree with this comment. Therefore, we have made changes in the text.

Sleep behavior is a prime example of how external factors like parental bedtime setting, cognitive behavioral therapy for insomnia (CBT-I) and a practice of good ‘sleep hygiene’ can modify children's and adolescents' physiological and developmental needs, leading to important outcomes for cognition, emotion regulation, motivation, and mental health [10]

Comment

Method

In the supplementary document there are questions about parents’ sleep, but this is not mentioned in the method section. Please give rationale why this was measured. What was the basis for all these questions posed to students? Which sleep studies?

Response: Thank you for pointing this out. We agree with this comment. Therefore, we have made changes in the text (introduction and methods) and have added references.

Nilsen S, Bergström M, Sivertsen B, Stormark K, Hysing M Sleep in adolescence: Considering family structure and family complexity. J. Marriage Fam. 2022;84:1152–1174. https://doi.org/10.1111/jomf.12844

Jeon, E.; Kim, N. Correspondence between Parents’ and Adolescents’ Sleep Duration. Int. J. Environ. Res. Public Health 2022, 19, 1034. https://doi.org/10.3390/ ijerph19031034

Comment

The measure of academic achievement was very crude and subjective. This needs to be highlighted in the limitations of the study.

Response: Thank you for pointing this out. We agree with this comment. Therefore, we have made changes in the text.

Comment

Results

I do not think it is precise or correct to call social habits behaviors like drinking or smoking. Please change it as it is very misleading. The same applies to the Discussion.

Response: Thank you for pointing this out. We agree with this comment. Therefore, we have made changes in the text. We used the term health habits instead throughout . The changes are highlighted.

Comment

What is the association between sleep duration in the week and on weekends?

Response: Thank you for pointing this out. We agree with this comment. Therefore, we have made changes in the text.  The changes are highlighted.

According to the results, students slept significantly longer on weekends compared to weekdays. Regardless of their level of education, students slept for a median of 7.0 hours (with an interquartile range of 6-7 hours) during weekdays, whereas on weekends, they slept for a median of 8.0 hours (IQR 7-9 hours, p<0.001). These findings are also presented in a table (2 ) in the revised manuscript as per the suggestions of reviewer 2.

Comment

In the section on Sleep quality a figure x is mentioned in the text. I assume it is a mistake.

Response:Thank you for pointing this out. We agree with the reviewer that Figure X was inserted by mistake. We have corrected this point in the revised manuscript.

Comment

Discussion

Page 7 – what is meant by environmental demands? In this context authors mean, school demands, socializing and I think. Please refine it.

Response: Thank you for pointing this out. We agree with this comment. Therefore, we have made changes in the text.

Comment

Can napping be more common in country like Greece?

Response: Thank you for pointing this out. According to the literature, napping is common among a large number of regions. Mediterranean region, Middle East, Spain, Italy and there are studies indicating that in the UK and US a part of the population have a nap. In addition, it seems to be a genetic predisposition for napping. There is no study to our knowledge indicating that napping is more common in Greece comparing to other Mediterranean countries

Comment

Please elaborate more on the link between parents' and their children's sleep.

Response: Thank you for pointing this out. We agree with this comment. Therefore, we have made changes in the text in several sections.

Comment

Please elaborate on the limitations of your measures of all behaviors - it was self-report, plus measures of alcohol and smoking were very crude. Why wasn't any validated health questionnaire used in the study?

Response: Thank you for pointing this out. We agree with this comment. Therefore, we have made changes in the limitations section

The questionnaire used was made on google platform for the needs of this study and not a validated scale already existed in the literature. The aim was to cover all the aspects of interest without using many different scales, which would eliminate the answering rates among school students due to long answering time. It is of great interest that the participation rate between students was almost 100%. All health-associated behaviors were self-reported and data about alcohol and smoking were crude. As a measure of academic achievement, we used only the school grades at the moment of participation in the study and this may not reflect the overall academic performance of a student. The study was conducted on Crete, which is the biggest island in Greece, but it is a specific region and the generalization of the results is limited. Further studies are needed to be done to better evaluated the sleep habits and health related behaviors among adolescents, as well as the effect of specific interventions in order to improve their sleep and wellbeing

Comments on the Quality of English Language

Minor editing of English language required.

Response: Thank you for pointing this out. We have made changes in the text.

Reviewer 2 Report

Comments and Suggestions for Authors

Dear authors,

Your study uncovered that the majority of adolescents in the Crete region experience sleep quality issues and deprivation. As adolescents age, their sleep duration decreases, with no significant differences observed between genders. Furthermore, the research showed a positive correlation between good sleep habits and academic success, as well as the avoidance of risky behaviors among these adolescents. The study's results underscore the importance of promoting sleep hygiene to improve academic performance and reduce risky behaviors among adolescents.

However, I believe the paper has some shortcomings, which I would like to mention:

-     In the context of the topic addressed, I consider the title incomplete. It would be beneficial for the title to be completed with the connection between sleep habits and social behavior, as well as academic performance of the subjects.

-     In the Keywords section, I believe academic performance and social habits of the subjects should be included.

- In the Introduction chapter, lines 38-39 "...in some areas of the world.", what are those areas and what sleep values are recorded there? Please specify! Also, I believe this chapter should be supplemented with data and studies on the relationship between positive social habits and sleep (e.g., physical activity) and negative social habits and sleep (e.g., alcohol, smoking, coffee). I am confident that such studies exist and should be mentioned in this context.

-  In Participants and methods, the registration dates of the study approvals are not specified. Additionally, I believe inclusion and exclusion criteria for the study should be presented. Did all students from the school participate?

-     You mentioned that the questionnaire used has 26 items, but in the supplementary material provided, there are 28 items. Is this a writing mistake or did you intentionally remove the 2 items from the analysis? Additionally, is the questionnaire a standard one? Is it adopted from other studies? Is it designed by you? Has it been validated? I believe these aspects should be specified!

-     In the titles of Figures 2 and 3, statistical data and interpretations are provided. Why didn't you interpret them separately and provide informational support through tables? Also, you presented data for parents without interpreting them.

-    In 3.2 Social Habits, you did not mention anything about physical activity, although it appears in the table. Please interpret the data about this!

-     Line 168 - figure X? Did you forget to present a figure? Please do so!

-     In general, I believe all these results should be presented not only by gender but also by age groups, to capture the behavioral trend of the subjects dynamically!

-    In the Discussion section, please present numerical data from the study compared to those taken from the sources studied. When you say "previous studies," "a series of studies," "some studies," please mention the sources. If there is only one, then mention and specify exactly which study it is. Also, when you mention "...for each of age group", please present the data obtained from your study (at 13 years old, 14 years old, etc.), and for each statement you make, please mention the sources from which it is derived.

-  Lines 226-231 - what you mention is true, but I believe these statements are unrelated to the topic at hand. You have no measurements or correlations of this kind... I suggest you remove this fragment!

-   In the limitations section, it should be added that the study was conducted in a specific region, Crete, not in southern Greece as the title suggests, and as such, the generalizability of the results is limited.

Author Response

Response to reviewer 2

Thank you very much for taking the time to review this manuscript. Please find the detailed point to point response below and the corresponding revisions/corrections highlighted in the re-submitted files.

-     In the context of the topic addressed, I consider the title incomplete. It would be beneficial for the title to be completed with the connection between sleep habits and social behavior, as well as academic performance of the subjects.

Response: Thank you for pointing this out. We agree with this comment. Therefore, we have made changes in the title.

-     In the Keywords section, I believe academic performance and social habits of the subjects should be included.

Response: Thank you for pointing this out. We agree with this comment. Therefore, we have made changes in the text.

- In the Introduction chapter, lines 38-39 "...in some areas of the world.", what are those areas and what sleep values are recorded there? Please specify! Also, I believe this chapter should be supplemented with data and studies on the relationship between positive social habits and sleep (e.g., physical activity) and negative social habits and sleep (e.g., alcohol, smoking, coffee). I am confident that such studies exist and should be mentioned in this context.

Response: Thank you for pointing this out. We agree with this comment. Therefore, we have made changes in the text and we have added references.

-  In Participants and methods, the registration dates of the study approvals are not specified. Additionally, I believe inclusion and exclusion criteria for the study should be presented. Did all students from the school participate?

Response: Thank you for pointing this out. We agree with this comment. Therefore, we have made changes in the text.

 We had no exclusion criteria except negative parental consent.  All students of the school participated in the study.

-     You mentioned that the questionnaire used has 26 items, but in the supplementary material provided, there are 28 items. Is this a writing mistake or did you intentionally remove the 2 items from the analysis? Additionally, is the questionnaire a standard one? Is it adopted from other studies? Is it designed by you? Has it been validated? I believe these aspects should be specified!

Response: Thank you for pointing this out. We agree with this comment. Therefore, we have made changes in the text. The correct number is 28 and 26 was written by mistake. The questionnaire was not a standard one. We addressed the limitations in the text:

The questionnaire used was made on google platform  for the needs of this study and not a validated scale already existed in the literature. The aim was  to cover all the aspects of interest without using many different scales which would eliminate the answering rates among school students due to long answering time. It is of great interest that  the participation rate between students was almost 100%.

-     In the titles of Figures 2 and 3, statistical data and interpretations are provided. Why didn't you interpret them separately and provide informational support through tables? Also, you presented data for parents without interpreting them.

Response

We agree with the reviewer, and in the revised manuscript, we present the data for students  and student’s parents sleep duration during weekdays and weekends  in the  table 2   and sleep duration per class in table 3

-    In 3.2 Social Habits, you did not mention anything about physical activity, although it appears in the table. Please interpret the data about this!

Response

 We have taken into consideration the reviewer's feedback and made necessary changes in the revised manuscript. In the results section, we have added a sentence which interprets the physical activity data. << A majority of students (54.3%) reported exercising often, with boys exercising at a higher proportion (58.4%) than girls (50.0%) (p=0.046).

-     Line 168 - figure X? Did you forget to present a figure? Please do so!

Response

In the revised manuscript, we have deleted Figure X as it has been inserted by mistake.

-     In general, I believe all these results should be presented not only by gender but also by age groups, to capture the behavioral trend of the subjects dynamically!

Response

We agree with the reviewer and in the revised manuscript we have added a table in supplement  ( Table S3) presenting health  habits data by age. We also added  a relative sentence in the results section.

-    In the Discussion section, please present numerical data from the study compared to those taken from the sources studied. When you say "previous studies," "a series of studies," "some studies," please mention the sources. If there is only one, then mention and specify exactly which study it is. Also, when you mention "...for each of age group", please present the data obtained from your study (at 13 years old, 14 years old, etc.), and for each statement you make, please mention the sources from which it is derived.

Response: Thank you for pointing this out. We agree with this comment. Therefore, we have made changes in the text.

-  Lines 226-231 - what you mention is true, but I believe these statements are unrelated to the topic at hand. You have no measurements or correlations of this kind... I suggest you remove this fragment!

Response: Thank you for pointing this out. We agree with this comment. Therefore, we have made changes in the text.

-   In the limitations section, it should be added that the study was conducted in a specific region, Crete, not in southern Greece as the title suggests, and as such, the generalizability of the results is limited.

Response: Thank you for pointing this out. We agree with this comment. Therefore, we have made changes in the limitations section.

Reviewer 3 Report

Comments and Suggestions for Authors

This study investigated the sleep patterns of secondary education students in Greece, and its association with school performance and social habits. Efforts to investigate students' sleeping habits are very important; however, the conclusions of this manuscript lack novelty. The results described here are all that have been found in previous studies. It is recommended that they be reorganized to better emphasize the originality of this study.

Specific issues to be addressed are as follows.

1. The introduction is not sufficiently organized in the first place There is an impression that the focus is not set on what the research was undertaken to clarify. This has led to inconsistency in the methods and results, including statistical analysis, and has created a poor impression of the paper as a whole.

2. What is it that you want to clarify in this study: grade and age differences? Gender differences? Differences in educational systems? Differences between weekdays and holidays?  Differences between parents and children?    In order to achieve the goal of assessing students' sleep patterns, it is better to start by sorting out these areas.

3. The author describes the research items in the supplemental file and omits much of the description of those items in the text, but is that procedure really appropriate? In order to make the paper easy to understand and read, it is necessary to learn how to write appropriately.

4. The Abstract should include the fact that it is an online survey, the type of scales used, and the analysis method conducted.

Author Response

Response to reviewer 3

Thank you very much for taking the time to review this manuscript. Please find the detailed point to point response below and the corresponding revisions/corrections highlighted in the re-submitted files.

  1. The introduction is not sufficiently organized in the first place. There is an impression that the focus is not set on what the research was undertaken to clarify. This has led to inconsistency in the methods and results, including statistical analysis, and has created a poor impression of the paper as a whole.

Response: Thank you for pointing this out. We agree with this comment. Therefore, we have made changes in the text.

  1. What is it that you want to clarify in this study: grade and age differences? Gender differences? Differences in educational systems? Differences between weekdays and holidays?  Differences between parents and children? In order to achieve the goal of assessing students' sleep patterns, it is better to start by sorting out these areas.

Response: Thank you for pointing this out. We want to evaluate the sleep habits and health behavior of the adolescents and how they are correlated with their academic performance. Sleep in adolescents is affected by all these factors.

Sleep Med Rev. 2021 June ; 57: 101425. doi:10.1016/j.smrv.2021.101425

Tarokh L, Saletin JM, Carskadon MA. Sleep in adolescence: Physiology, cognition and mental health. Neurosci Biobehav Rev. 2016 Nov;70:182-188. doi: 10.1016/j.neubiorev.2016.08.008. Epub 2016 Aug 13. PMID: 27531236; PMCID: PMC5074885.

  1. The author describes the research items in the supplemental file and omits much of the description of those items in the text, but is that procedure really appropriate? In order to make the paper easy to understand and read, it is necessary to learn how to write appropriately.

Response: We want to thank the reviewer for the comment.  However, in the original manuscript, in the section on sleep habits, we have described that tables S1 and S2  presents Pairwise comparisons of sleep duration  during weekdays and weekends between the six classes

  1. The Abstract should include the fact that it is an online survey, the type of scales used, and the analysis method conducted.

Response: We want to thank the reviewer for the comment. We agree with this comment. We made a few corrections but unfortunately the abstract has a 200 word limitation and we are not allowed to include every detail. We mention the requested information in the text.

Round 2

Reviewer 2 Report

Comments and Suggestions for Authors

Dear Authors,

Thank you for considering my suggestions, and I believe the work has been greatly improved.

I understand that the questionnaire was not a standard one but was designed by you. Therefore, it should be validated by calculating the Cronbach's alpha index! I believe this will add value and can be used by other researchers in their work as well!

Author Response

Dear Reviewer,

Thank you for your valuable remark. The Cronbach’s alpha coefficient was used especially in questionnaires where a common scale of items measures was used. As an example questionnaires used a Liker scale of n-responses could be used. In our case our items (questions) were not of the same scale measuring different traits without unidimensionality. Additionally, Cronbach’s alpha was used as a test of internal consistency not validity of a scale (Sijtsma & Pfadt, 2021). Supporting our sentence of the non-appropriateness of Cronbach’s alpha use we mention the example of Bland & Altman in 1997 publication were data of the same scale (Bland & Altman, 1997) and a useful article of misuses of this coefficient (Tavakol & Dennick, 2011).

Bland, J. M., & Altman, D. G. (1997). Cronbach's alpha. BMJ (Clinical research ed.), 314(7080), 572. https://doi.org/10.1136/bmj.314.7080.572

Tavakol M, Dennick R. Making sense of Cronbach's alpha. Int J Med Educ. 2011 Jun 27;2:53-55. doi: 10.5116/ijme.4dfb.8dfd. PMID: 28029643; PMCID: PMC4205511.

Sijtsma K, Pfadt JM. Part II: On the Use, the Misuse, and the Very Limited Usefulness of Cronbach's Alpha: Discussing Lower Bounds and Correlated Errors. Psychometrika. 2021 Dec;86(4):843-860. doi: 10.1007/s11336-021-09789-8. Epub 2021 Aug 13. PMID: 34387809; PMCID: PMC8636457.

Reviewer 3 Report

Comments and Suggestions for Authors

1.        The authors state that they would like to "assess the sleep habits and health behaviors of youth and examine how they correlate with academic performance," however from the revised manuscript, this does not seem to be necessary. The author's wishes can be fulfilled immediately through a proper search of previous studies. Nevertheless, if the authors wish to publish their research as a paper, they need to state the theoretical basis that what they want to know cannot be explained by the findings of previous studies. That rationale is what reveals the novelty of your research. From the revised parts of the revised manuscript, the above rationale remains unclear and is unlikely to be novel. At the same time, the significance of this study is also unclear because there is no description in the revised manuscript as to why it is necessary to study the sleep patterns of adolescent students in Crete, Greece. Furthermore, the paper is developed without providing any theoretical basis for its examination by gender, age, educational level, and class level. Since there is no explanation as to why such detailed stratification is necessary and what might be revealed by examining such stratification, it is very difficult to understand. Also, there is no research hypothesis, which also makes this study difficult to understand.

2.        The Abstract should include the fact that the type of scales used, and the analysis method conducted.  The 200-word limit on abstracts is not a reason why abstracts cannot be revised. An abstract that lacks adequate description detracts from the value of the study. The title and abstract are the face of the study and should be examined for content. Unnecessary sections should be deleted, and appropriate information added.

3.        The authors are content to submit the online questionnaire as Supplementary, but is that really the proper response? It is difficult for the reader to review the Supplementary each time he or she wants to check on the variables used in the analysis. Wouldn't it be better to provide a brief explanation of what options were used for the major variables and what responses were requested?

4.        The logistic regression model is used; however, the outcome, poor academic performance, was not defined.

5.        In the first paragraph of the discussion, there is a statement regarding gender differences. What are the possible reasons for some of the differences noted in the discussion from previous studies? Please explain the differences with proper rationale.

Comments on the Quality of English Language

Please check carefully for typographical errors before submission.

Author Response

Dear reviewer

Thank you for your valuable remarks.
